# *DIS3*: The Enigmatic Gene in Multiple Myeloma

**DOI:** 10.3390/ijms24044079

**Published:** 2023-02-17

**Authors:** Yasuyo Ohguchi, Hiroto Ohguchi

**Affiliations:** Division of Disease Epigenetics, Institute of Resource Development and Analysis, Kumamoto University, 2-2-1 Honjo, Chuo-ku, Kumamoto 860-0811, Japan

**Keywords:** DIS3, multiple myeloma, genetic events, RNA exosome, hematopoiesis

## Abstract

Recent studies have revealed the genetic aberrations involved in the initiation and progression of various cancers, including multiple myeloma (MM), via next-generation sequencing analysis. Notably, *DIS3* mutations have been identified in approximately 10% of patients with MM. Moreover, deletions of the long arm of chromosome 13, that includes *DIS3*, are present in approximately 40% of patients with MM. Regardless of the high incidence of *DIS3* mutations and deletions, their contribution to the pathogenesis of MM has not yet been determined. Herein, we summarize the molecular and physiological functions of DIS3, focusing on hematopoiesis, and discuss the characteristics and potential roles of *DIS3* mutations in MM. Recent findings highlight the essential roles of DIS3 in RNA homeostasis and normal hematopoiesis and suggest that the reduced activity of DIS3 may be involved in myelomagenesis by increasing genome instability.

## 1. Introduction

Multiple myeloma (MM) is an incurable plasma cell malignancy that accounts for approximately 10% of all hematologic cancers [1]. Development and progression of MM are characterized by clonal evolution, which confers tumor plasticity and drug resistance to tumor cells [1,2,3]. Therefore, it is important to understand the underlying genetic abnormalities and mechanisms of clonal evolution to identify potential therapeutic targets for MM. Recent advances in genome sequencing technologies have revealed the genetic driver events in MM. Gene mutations activating the mitogen-activated protein kinase pathway (including those in *KRAS*, *NRAS,* and *BRAF*) and the nuclear factor-κB pathway (including those in *TRAF3*, *LTB*, and *CYLD*) have been identified in approximately 40 and 20% of patients with MM, respectively [1,2,3,4]. Genes related to the DNA repair pathway (including *TP53*, *ATM*, and *ATR*), G1/S cell cycle transition (including *CCND1* and *RB1*), and epigenetic regulation (including *HIST1H1E*, *KMT2C*, and *KDM6A*) have also been reported to be recurrently mutated in MM cases, confirming their importance in the pathogenesis of MM [1,2,3,4]. It should be noted that, in addition to these relatively common mutations in cancers, MM-specific mutations involving *DIS3* and *FAM46C* (also known as *TENT5C*) have been identified in about 10% of patients with MM each [1,2,3,4]. Furthermore, deletion of the chromosomal region containing *DIS3* or *FAM46C* has also been observed in around 40 or 30% of patients with MM, respectively [2,3], suggesting the pathological importance of these genes in MM. Although recent studies have revealed the molecular functions of FAM46C and its tumor-suppressive roles in MM [5,6,7,8,9,10], the biological significance of *DIS3* mutations in MM remains poorly understood. In this review, we describe the essential roles of DIS3 in RNA homeostasis and hematopoiesis and outline the characteristics and clinical impact of *DIS3* mutations in MM. We further discuss the pathological potential of *DIS3* mutations in MM. A recent review has summarized the potential roles of FAM46C in MM [11].

## 2. DIS3: A Catalytic Subunit of the RNA Exosome

### 2.1. Structure and Functions of the RNA Exosome

RNA species play essential roles in mediating and regulating gene expression; therefore, their qualitative and quantitative control is vital for cellular homeostasis [12,13,14,15]. The RNA exosome is a multiprotein complex that possesses ribonuclease (RNase) activity and is a key player in executing qualitative and quantitative control of RNAs via both degradative and modification reactions [12,13,14,15]. The RNA exosome degrades RNAs when they complete their roles (turnover) or when they turn out to be defective (surveillance). In addition to RNA degradation, the RNA exosome is involved in the 3′-end processing of precursor RNAs (maturation) [12,13,14,15].

The RNA exosome consists of a barrel-shaped nine-subunit core (EXO9) and two catalytic subunits (Figure 1) [12,13,14,15,16]. EXO9 is composed of a hexameric ring (EXOSC4, EXOSC5, EXOSC6, EXOSC7, EXOSC8, and EXOSC9) that surrounds the central channel of the exosome and a trimeric cap (EXOSC1, EXOSC2, and EXOSC3) that resides on the top of the hexameric ring and carries the S1 and KH types of RNA-binding domains [12,13,14,15,16]. EXO9 is required for RNA presentation to catalytic subunits and association with other cofactors, such as the Trf4–Air2–Mtr4 polyadenylation (TRAMP) and nuclear exosome-targeting (NEXT) complexes, which support the exosome functions [13,14,15,17,18]. The catalytic subunit, DIS3, or its homologue, DIS3L, is anchored at the bottom of the hexameric ring of EXO9 in a mutually exclusive manner [13,14,15,16,19,20]. The other catalytic subunit, EXOSC10, interacts with the cap side of EXO9 [13,14,15,16]. Components of the exosome complex differ between the cellular compartments. In human cells, EXO9 associates with both DIS3 and EXOSC10 in the nucleus and only with EXOSC10 in the nucleolus [19,20]. DIS3L is present only in the cytoplasm; EXO9 interacts with DIS3 or DIS3L in addition to EXOSC10 in the cytoplasm [19,20]. Although their functional significance remains unknown, the presence of different exosome isoforms suggests isoform-specific functions in distinct cellular compartments [13,14]. For RNA degradation, substrates enter through a pore at the center of the trimeric cap and are threaded through a central channel to access the catalytic subunit, DIS3/DIS3L (Figure 1) [13,14,16,21]. Alternatively, RNAs reach the active center of EXOSC10 via undetermined routes and are degraded by EXOSC10 [14,22,23]. In contrast, RNAs that undergo maturation are proposed to be directly targeted by DIS3 without being threaded through a central channel [24,25]. Although the mechanisms by which the exosome differently targets RNAs remain unknown, recent studies suggest that the properties of RNAs and the cofactors associated with the exosome determine the way the exosome degrades or processes RNAs [14,15].

The RNA exosome targets a large variety of RNAs in both the nucleus and cytoplasm (Table 1). In the nucleus, the exosome removes unstable pervasive transcripts, such as promotor upstream transcripts (PROMPTs) and enhancer RNAs (eRNAs) to prevent their accumulation at an inappropriate level [12,13,14,15]. It also eliminates all classes of defective stable RNAs, including ribosomal RNAs (rRNAs), transfer RNAs (tRNAs), messenger RNAs (mRNAs), small nuclear RNAs (snRNAs), and small nucleolar RNAs (snoRNAs), which are improperly processed and harmful [12,13,14,15]. Furthermore, the nuclear exosome participates in 3′-end processing for the maturation of rRNAs, snRNAs, and snoRNAs [12,13,14,15]. In the cytoplasm, the exosome is engaged in the turnover of normal mRNAs and degradation of aberrant mRNAs via nonsense-mediated, nonstop, and no-go decay [12,13,14,15]. The cytoplasmic exosome also eliminates mRNAs harboring AU-rich sequence elements (AREs) that encode proteins for which rapid turnover is crucial [12,13,14,15]. Thus, the RNA exosome plays a central role in RNA homeostasis, thereby maintaining proper cellular functions.

Defects in the exosome subunits are implicated in several diseases. *DIS3* mutations have been observed not only in MM but also in acute myeloid leukemia [26]. Deletion of the *DIS3* locus has also been reported in chronic lymphocytic leukemia [27]. *EXOSC3* and *EXOSC8* mutations are associated with neurodegenerative disease, pontocerebellar hypoplasia type 1b and type 1c, respectively [28]. *EXOSC2* mutations have been identified in cases with a novel syndrome, which represents various phenotypes, including retinitis pigmentosa, premature aging, and mild intellectual disability [29].

### 2.2. Structure and Molecular Functions of DIS3

DIS3 is a member of the RNase II/R family and well conserved from yeast to humans. It possesses conserved motif and domains related to its RNase function (Figure 2) [13,19,30]. The PilT N-terminal (PIN) and RNB domains contain catalytic cores and are responsible for endo- and exonucleolytic activity, respectively [31,32,33,34]. Both endo- and exonuclease activities of DIS3 are required for efficient RNA degradation and processing [32,33,34,35]. Two tandem cold-shock domains (CSD1 and CSD2) and the S1 domain confer substrate-binding capacity [36]. The Cysteine-Rich with three cysteines (CR3) motif is involved in the interaction with EXO9, possibly by affecting the conformation of the residues that bind to EXOSC4 [37]. The CR3 motif also supports the endonuclease activity of the PIN domain by physically interacting with its active site [37]. DIS3L possesses all conserved motif and domains but lacks two catalytic residues within the PIN domain, resulting in defect in its endonuclease activity [19]. In humans, there is another DIS3 homologue, DIS3L2. However, DIS3L2 lacks the CR3 motif and PIN domain and functions independently of the RNA exosome [19,38].

Although the RNA exosome targets various types of RNAs, recent studies have underscored the importance of DIS3 in the removal of pervasive transcripts [35,39]. Pervasive transcripts are unstable RNAs that are barely detectable under normal conditions due to exosome-mediated rapid degradation [40,41]. In human cells, DIS3 depletion results in the accumulation of pervasive transcripts, including PROMPTs, eRNAs, and products of premature cleavage and polyadenylation (PCPA), but EXOSC10 depletion does not affect the levels of pervasive transcripts [35,39]. These findings suggest that DIS3, but not EXOSC10, removes pervasive transcripts in a nonredundant manner. Notably, one study reported the substantial accumulation of substrates within 60 min of DIS3 depletion, highlighting the dynamic DIS3 function [39]. In another study, DIS3 deficiency resulted in altered expression of approximately 50% of mRNAs; however, this alteration seems to be a secondary effect of the accumulation of noncoding RNAs because very little correlation was observed between the upregulated and DIS3-bound mRNAs [35]. DIS3 is also involved in the degradation of the shorter form of NEAT1 (NEAT1.1) [35]. NEAT1 is a long noncoding (lnc)-RNA that forms paraspeckles in the nucleus [42]. Indeed, *DIS3* deficiency has been shown to increase the number and volume of paraspeckles; however, the biological significance of this phenomenon remains to be determined [35]. Finally, DIS3 is engaged in the processing of snoRNAs, whereas EXOSC10 is involved in the degradation of mature snoRNAs, further suggesting the functional difference between the two exosome enzymes [35].

## 3. Physiological Functions of DIS3 in Hematopoiesis

### 3.1. DIS3 Functions in B Cell Development

Although DIS3 functions in hematopoiesis remain largely elusive, recent studies have revealed its functions in B cell development [43,44]. To generate high-affinity antibodies against various foreign antigens, differentiating B cells undergo immunoglobulin (*Ig*) gene rearrangements, including V(D)J recombination, somatic hypermutation (SHM), and class-switch recombination (CSR) [45]. V(D)J recombination occurs during the pro-B/pre-B transition in the bone marrow, and SHM and CSR occur in activated B cells in the germinal center (GC) after B cells move to secondary lymphoid organs (Figure 3) [45]. Importantly, RNA exosome components, including *Dis3*, are highly expressed in pro-B/pre-B and GC B cells, which is associated with the timing of *Ig* gene rearrangements (Figure 3) [44]. Indeed, DIS3 is involved in these processes. Laffleur et al. generated early B cell-specific *Dis3* conditional knockout mice using *Mb1*^cre^ mice; *DIS3*-deficient pro-B cells exhibited defects in V(D)J recombination and pre-B cell receptor signaling, resulting in the failure of the pro-B to pre-B transition [44]. They suggested that an aberrant accumulation of pervasive noncoding RNAs at the *Ig* genes due to *DIS3* deficiency leads to a defect in V(D)J recombination, presumably by impeding the access to V(D)J recombination-related components, including RAG recombinases, as the expression of these components is not altered in *DIS3*-deficient pro-B cells [44]. Laffleur et al. also generated an activated B cell-specific *Dis3* knockout mouse model and showed that DIS3 is required for proper SHM and CSR [43]. *DIS3*-deficient activated B cells exhibit increased DNA:RNA hybrids in the V(D)J regions, which are detected by a DNA:RNA hybrid immunoprecipitation-sequencing approach, resulting in limited accessibility of activation-induced cytidine deaminase (AID) to antisense strand DNA and distinct patterns of SHM [43]. *DIS3*-deficient activated B cells also exhibit the accumulation of DNA:RNA hybrids in the *Igh* topologically associating domains (TADs), leading to reduced CTCF/cohesin binding to these regions and decreased *Igh* TAD interactions, subsequently impeding CSR [43]. These studies demonstrate that DIS3-sensitive noncoding transcripts efficiently generate the DNA:RNA hybrid in *DIS3*-deficient B cells and that DIS3 is essential for *Ig* gene rearrangements and B cell maturation via the removal of pervasive noncoding transcripts (Figure 3). An open question is whether DIS3 function that prevents the formation of DNA:RNA hybrids is B cell-specific or not. In yeast, DIS3 dysfunction does not increase DNA:RNA hybrids [46], suggesting tissue/species-specific DIS3 function. Further studies are required to address this issue.

### 3.2. DIS3 Functions in Erythropoiesis

DIS3 has also been shown to be involved in erythroid homeostasis. During erythropoiesis, GATA1 and Foxo3 strongly repress the expression of RNA exosome components, including *DIS3* [47]. Thus, DIS3 is not required for erythroid maturation; however, shRNA-mediated DIS3 depletion prior to GATA1-driven repression in hematopoietic progenitor cells compromises the formation of burst-forming unit-erythroid (BFU-E) and colony-forming unit-erythroid (CFU-E) in in vitro systems [48]. Mechanistic analysis suggests that DIS3 protects erythroid precursor cells from DNA damage-induced apoptosis, in part, via c-Kit signaling [48]. Thus, DIS3 contributes to the survival of erythroid precursor cells before GATA1-mediated differentiation, thereby maintaining erythropoiesis [48]. DIS3 may also play roles in other hematopoietic cells, and this should be explored in future studies.

## 4. Characteristics and Clinical Impact of *DIS3* Mutations in MM

### 4.1. Characteristics of DIS3 Mutations in MM

*DIS3* is located on chromosome 13q, and this region is heterozygously deleted in around 40% of MM cases [2]. In addition to the heterozygous deletion of this gene, Chapman et al. identified *DIS3* mutations in MM using whole-genome and whole-exome sequencing in 2011 [49]. Subsequent studies have confirmed that *DIS3* mutations are recurrent and present in approximately 10% of patients with MM [4,50,51,52,53,54]. Germline mutations in *DIS3* have also been reported in familial MM, suggesting the pathological relevance of *DIS3* mutations in MM [55]. In the Multiple Myeloma Research Foundation (MMRF) CoMMpass cohort, which included 930 patients with MM, the variant allele frequency (VAF) ranged from 5.3 to 100% (mean: 48%; median: 43%), indicating the presence of *DIS3* mutations in both major and minor subclones [54]. Most of the *DIS3* mutations observed in MM are missense mutations, and nonsense mutations are barely observed [4,49,50,51,52,53,54]. Notably, *DIS3* mutations are primarily present in the catalytic domains of this protein; about 70% of the mutations are located in the RNB domain, and about 10–20% are located in the PIN domain, suggesting the functional relevance of these mutations [52,53,54]. Indeed, the majority of *DIS3* mutations are located in highly conserved residues across species [52,53,54]. Mutational hotspots include arginine at position 780 (R780), aspartic acid at position 488 (D488), and aspartic acid at position 479 (D479), all of which are within the RNB domain (Figure 2) [4,52,53,54]. Based on the analysis of DIS3 in yeast, it has been shown that R780 is engaged in RNA binding, while D488 and D479 are engaged in magnesium ion binding at the active site, which is required for the RNA degradation activity of this enzyme [31,56]. Although the mutational profile of *DIS3* (high rate of missense mutations at the same codons and low rate of transcription-terminating mutations) suggests that it functions as an oncogene [4], it is proposed to function as a tumor suppressor gene due to the following reasons: (1) MM-related *DIS3* mutations, such as R780K, exhibit reduced enzymatic activity in a biochemical assay [57], (2) *DIS3* mutations are frequently accompanied by loss of heterozygosity via 13q deletion [4,49,50,51,52,53,54], and (3) the transcriptional profiles in samples from MM patients with *DIS3* mutations are consistent with the malfunction of the RNA exosome [53,54]. Recent mechanistic studies further support the notion that DIS3 is a tumor suppressor, which we discuss later in this review.

### 4.2. Correlations of DIS3 Mutations with Cytogenetic Abnormalities and Other Mutations in MM

*DIS3* mutations are more frequently observed in nonhyperdiploid MM patients than in hyperdiploid MM patients [4,50,51,52,53,54]. In agreement with the fact that *IGH* translocations are common genetic events in nonhyperdiploid MM [1,2,3], there is a strong association between *DIS3* mutations and *IGH* translocations, such as t(4;14), t(14;16), and t(11;14) [4,50,51,52,53,54]. There is also a significant co-occurrence of *DIS3* mutations with 13q deletions, which results in biallelic events in *DIS3* [4,50,51,52,53,54]. In the MMRF CoMMpass cohort, 72% of patients with *DIS3* mutations exhibited 13q deletions [54]. Interestingly, only 31% of cases harboring hotspot *DIS3* mutations represented 13q deletions, whereas 93% of cases harboring nonhotspot mutations carried 13q deletions in this cohort [54]. Furthermore, it has been shown that hotspot mutations are never present at VAFs higher than 50% [53]. As described in the previous section, mutational hotspot sites are closely related to the enzymatic function of DIS3, and hotspot mutations cause severe impairment of enzymatic activity [31,56,57]. Thus, complete inactivation of this enzyme may be deleterious to cellular survival, whereas partial reduction of enzymatic activity may have a beneficial effect on the pathogenesis of MM. Indeed, the homozygous deletion of *DIS3* has not been reported in patients with MM. We and other groups also noted that after knocking out the *DIS3* gene in cell lines, homozygous-knockout clones could not be obtained, whereas heterozygous-knockout clones could (Y. Ohguchi and H. Ohguchi, unpublished observation) [57]. Regarding other cytogenetic alterations and mutations, a significant association between *DIS3* mutations and 1q21 gain or *BRAF* mutations has been reported [4,51,54]. In contrast, an inverse association with 1p22 loss has been observed [54]. Although a significant correlation between *DIS3* mutations and specific cytogenetic alterations has been determined, whether they collaborate in the development of MM remains elusive.

### 4.3. Association of DIS3 Mutations with Prognosis in MM

Three different cohort studies explored the impact of *DIS3* mutations on the prognosis of patients with MM [52,54,58]. In the Deutsche Studiengruppe Multiples Myelom (DSMM) XI study (*n* = 81), there was a trend toward a lower median overall survival (OS) in *DIS3*-mutated patients than in *DIS3*-wild type patients, although no statistical significance was observed (33 vs. 54 months, *p* = 0.138) [52]. In the MMRF CoMMpass study (n = 930), both OS (3-year OS rates: 65 vs. 79%, *p* = 0.039) and progression-free survival (PFS; median PFS: 800 vs. 1176 days, *p* = 0.021) were significantly shorter in *DIS3*-mutated patients than in *DIS3*-wild type patients [54]. Multivariate analysis further confirmed monoallelic and biallelic *DIS3* lesions as independent prognostic factors of poor OS (*p* = 0.01) and PFS (*p* = 0.014), respectively [54]. Therapeutic regimens did not affect the OS and PFS of *DIS3*-mutated patients in this cohort [54]. Shorter event-free survival (EFS) was also noted in *DIS3*-mutated patients than in *DIS3*-wild type patients (*p* = 0.008) in the total therapy trial cohort (*n* = 223) [58]. Multivariate analysis validated the independent association of *DIS3* mutations with worse EFS in that study (*p* < 0.001) [58]. These studies indicate that *DIS3* mutations negatively affect the prognosis of patients with MM.

### 4.4. Transcriptional Profiles of MM Patients with DIS3 Mutations

Transcriptome analysis of *DIS3*-mutated MM samples revealed a transcriptional signature associated with *DIS3* mutations [53,54,59]. Overall, *DIS3* mutations are associated with the upregulation of noncoding RNAs, especially lncRNAs, which presumably reflects the impaired RNA-degradative function of DIS3 [53,54]. Consistent with this observation, genes involved in RNA metabolism are downregulated in *DIS3*-mutated cases [54]. In contrast, genes involved in interferon signaling are upregulated in *DIS3*-mutated cases [53,54,59]. As discussed later, loss-of-function of DIS3 has been shown to lead to genome instability [43,59]. Genome instability stimulates the cGAS–STING pathway, thereby inducing an interferon response [60]. Thus, activation of the interferon signaling may reflect genome instability induced by *DIS3* mutations (Figure 4) [59]. Todoerti et al. also showed that genes involved in KRAS signaling, cell adhesion, and JAK-STAT signaling are upregulated in *DIS3*-mutated cases, suggesting involvement in myelomagenesis [54]. Regarding the clinical relevance of differentially expressed transcripts in *DIS3*-mutated cases, five lncRNAs (*AC015982.2*, *AL353807.2*, *AC013400.1*, *ASH1L-AS1*, and *AL445228.3*) are associated with a shorter OS, two of which (*AC015982.2* and *AL445228.3*) are also independent predictors of PFS, although future studies are required to determine the pathological roles of these lncRNAs in MM [54].

## 5. Pathological Functions of DIS3 Hypomorphism in MM

Although there is no firm evidence whether genetic alterations involving *DIS3* are actionable in myelomagenesis, recent studies have suggested the pathological significance of loss-of-function of DIS3. The *let-7* microRNA family is a tumor suppressor family that inhibits the translation of oncogenes, including *RAS* and *MYC* [61]. Segalla et al. showed that depletion of *DIS3* increases the expression of *LIN28B*, a negative regulator of *let-7*-maturation, thereby increasing the protein levels of RAS and MYC in MM cells (Figure 5) [62]. Thus, this study revealed a novel RAS and MYC activation mechanism in MM cells. Loss of DIS3 activity has also been implicated in genome instability. In *DIS3*-deficient B cells, aberrant DNA:RNA hybrids impair the architectural integrity of TADs by reducing CTCF/cohesin binding, leading to increased inter-TAD recombination (chromosomal translocations) (Figure 5) [43]. Thus, loss-of-function of DIS3 may be involved in MM-associated translocations. In fact, *DIS3* mutations are preferentially observed in MM patients with *IGH*-translocations [52,53,54]. *DIS3* deficiency-induced DNA:RNA hybrids also interfere with the recruitment of the homologous recombination machinery to double-strand breaks, thereby increasing DNA damage and mutational load in cancer cells, including MM cells (Figure 5) [59]. In agreement with this observation, MM patients with *DIS3* mutations exhibit a higher mutational burden than those without *DIS3* mutations [59]. Thus, *DIS3* deficiency may promote myelomagenesis by endowing driver mutations in MM cells. However, late B cell-specific *Dis3* knockout mice with a C57BL/6 background barely exhibited B cell malignancy over a two-year observation period (Y. Ohguchi and H. Ohguchi, unpublished observations), suggesting the involvement of additional oncogenic events in tumorigenesis. Snee et al. showed that, although loss of DIS3 activity impairs cell division, reduced DIS3 activity in concert with RAS activation enhances cell proliferation in *Drosophila* and murine B cell models, supporting this idea [63]. Further efforts are necessary to develop in vivo MM models harboring *DIS3* deletion/mutations to decipher the precise function of DIS3 hypomorphism in the pathogenesis of MM.

## 6. Conclusions and Future Perspectives

Recent progress in genomic sequencing have uncovered genomic complexity and recurrent genetic events in cancers, including MM. These findings have led to a paradigm shift in the understanding of cancers; however, the mechanisms by which these genetic abnormalities contribute to the pathogenesis of cancers have not been fully understood. *DIS3* mutations in MM are such genetic abnormalities, the pathological functions of which have not yet been defined. In this review, we have summarized the recent knowledge of DIS3 functions in hematopoiesis and *DIS3* mutations in MM. DIS3 is indispensable for *Ig* gene rearrangements and B cell maturation, and loss-of-function of DIS3 results in genome instability via the formation of an R-loop (DNA:RNA hybrids), which is involved in increased chromosomal translocations and mutational burden [43,59]. Multiple lines of evidence support the idea that *DIS3* mutations are loss-of-function mutations in MM; however, the biallelic loss of this gene is lethal [4,49,50,51,52,53,54,57]. Taken together, these findings suggest that *DIS3* mutations may be so-called mutator mutations [64], that is, reduced DIS3 activity caused by *DIS3* mutations may lead to MM-related translocations and mutations, but this hypothesis needs to be investigated further in future studies. Recent studies have also identified a significant association between *DIS3* mutations and other MM-related genetic events. An outstanding question is this: what is the pathological significance of this association? Co-occurrence of *DIS3* mutations with such genetic events may overcome the negative effect of DIS3 hypomorphism, thereby promoting MM. One study on *Drosophila* models reported that reduced DIS3 activity in combination with RAS activation stimulates cell growth, but reduced DIS3 activity alone impairs cell growth [63]. Further detailed studies are required to elucidate the collaborative mechanisms of these genetic events. These mechanistic insights will pave the way to novel therapeutic strategy to improve the outcome of patients with MM.

## Figures and Tables

**Figure 1 ijms-24-04079-f001:**
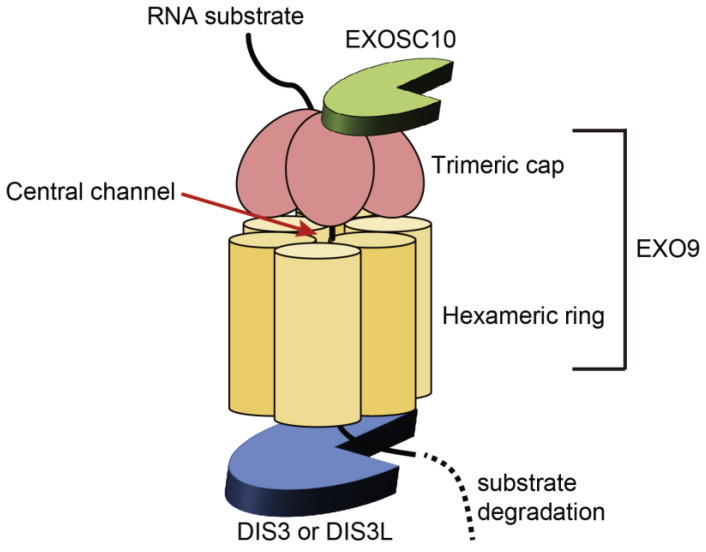
Structure of the RNA exosome. The RNA exosome comprises an enzymatically inactive, barrel-shaped nine-subunit core (EXO9) and two catalytic subunits (DIS3/DIS3L and EXOSC10). EXO9 includes a hexameric ring (EXOSC4, EXOSC5, EXOSC6, EXOSC7, EXOSC8, and EXOSC9) and a trimeric cap (EXOSC1, EXOSC2, and EXOSC3). The catalytic subunit, DIS3, or its homologue, DIS3L, interacts with EXO9 at the bottom of the hexameric ring in a mutually exclusive manner. The other catalytic subunit, EXOSC10, binds to the cap side of EXO9. RNA substrates are threaded through a central channel to the catalytic core of DIS3 or DIS3L for degradation.

**Figure 2 ijms-24-04079-f002:**
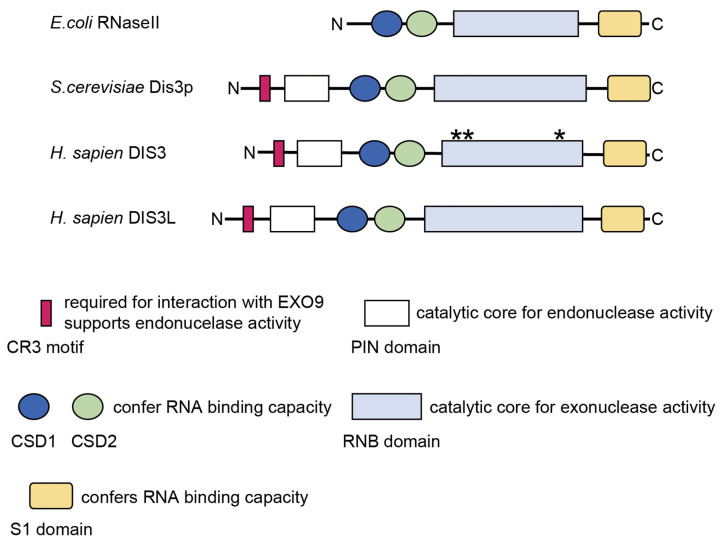
Motif and domains of the DIS3 protein family. Functions of each motif and domain are shown. PilT N-terminal (PIN) domain of DIS3L does not possess any endonuclease activity due to the lack of catalytic residues. Each of * indicates the position of the mutational hotspots in multiple myeloma, which are located in the RNB domain. CR3, Cysteine-Rich with three cysteines; CSD, cold-shock domain.

**Figure 3 ijms-24-04079-f003:**
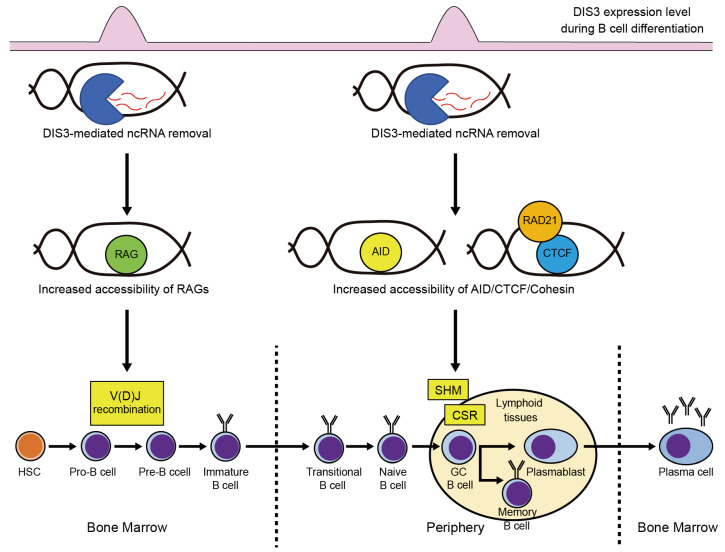
DIS3 is necessary for immunoglobulin gene rearrangements and B cell differentiation. Induced DIS3 removes noncoding RNAs (ncRNAs) to prevent the formation of DNA:RNA hybrids during immunoglobulin gene rearrangements, which increases the accessibility of RAG recombinases, activation-induced cytidine deaminase (AID), CTCF, and cohesin complex, leading to successful V(D)J recombination, somatic hypermutation (SHM), and class-switch recombination (CSR). HSC, hematopoietic stem cell; GCB cell, germinal center B cell.

**Figure 4 ijms-24-04079-f004:**
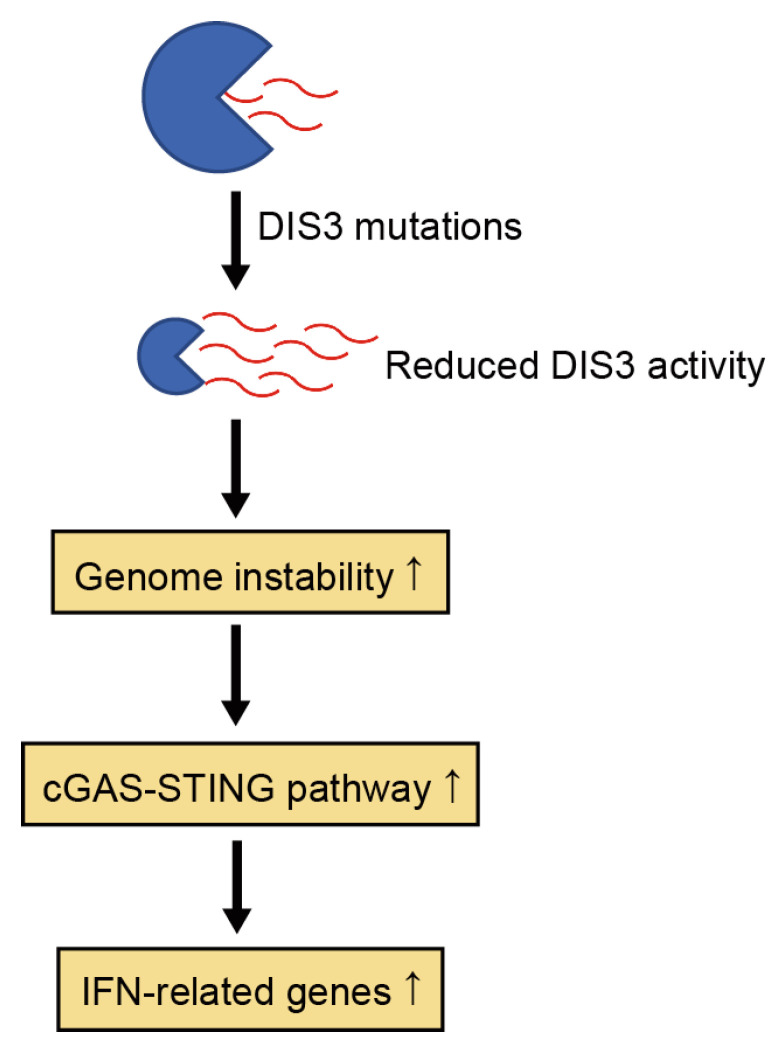
DIS3 mutations induce an interferon response. Reduced DIS3 activity increases genome instability, which may activate the cGAS-STING pathway, upregulating interferon (IFN)-related genes.

**Figure 5 ijms-24-04079-f005:**
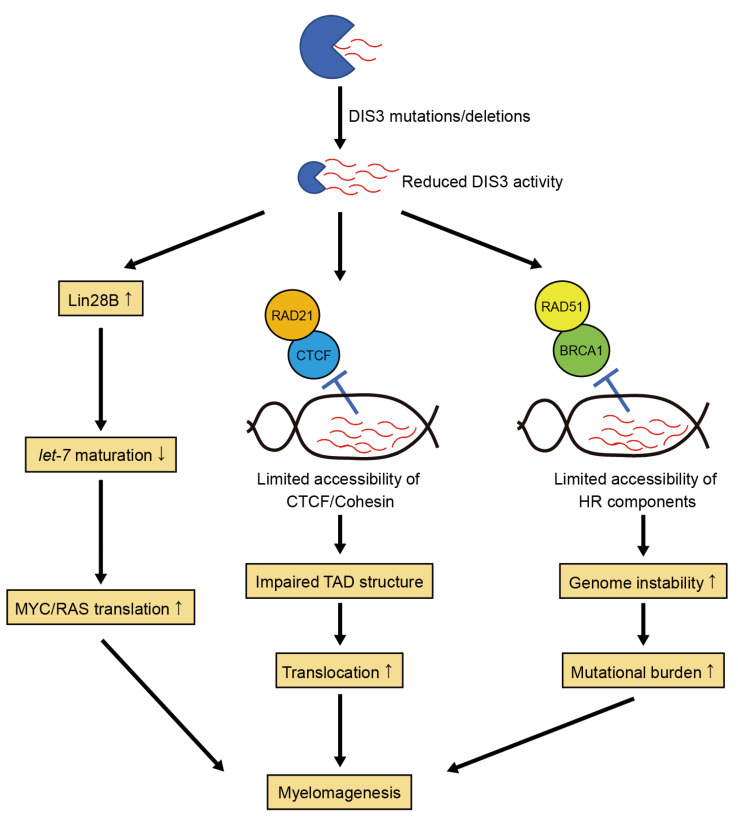
Potential hypomorphic functions of DIS3 in myelomagenesis. Reduced DIS3 activity increases Lin28B levels, which inhibits *let-7* maturation, leading to increased MYC and RAS protein levels. Reduced DIS3 activity also induces the formation of DNA:RNA hybrids, thereby limiting the accessibility of CTCF and cohesin complex, including RAD21. This impairs topologically associating domain (TAD) structure, resulting in chromosomal translocations. Aberrant DNA:RNA hybrids also limit the accessibility of homologous recombination (HR) components, including RAD51 and BRCA1, thereby increasing genome instability and mutational burden.

**Table 1 ijms-24-04079-t001:** Functions of the RNA exosome.

Location	Target RNAs	Exosome Function
Nucleus	Pervasive transcripts (PROMPTs and eRNAs)	Degradation
Nucleus	Defective RNAs (rRNAs, tRNAs, mRNAs, snRNAs, and snoRNAs)	Degradation (surveillance)
Nucleus	Precursor RNAs (rRNAs, snRNAs, and snoRNAs)	Processing (maturation)
Cytoplasm	Normal mRNAs	Degradation (turnover)
Cytoplasm	Defective mRNAs	Degradation (surveillance)
Cytoplasm	mRNAs harboring ARE	Degradation (ARE-mediated decay)

## Data Availability

Not applicable.

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
