# Peer review of "DIS3*: The Enigmatic Gene in Multiple Myeloma"

_ijms, 2023, doi:10.3390/ijms24044079_

Round 1
Reviewer 1 Report
This is an interesting review documenting up to date information on the role of DIS3 mutations in multiple myeloma. While author's primary focus is on molecular and physiological functions of DIS3 in hematopoiesis, they extend their explorations to role of DIS3 mutations to other cancers also. The review is well developed with a focus on as a catalytic subunit of the RNA exosome, DIS3 contributes to different biological processes and diseases. The review is supported by clear figures and descriptions. Overall very informative and well written review with suggestions for further work to provide mechanistic insights for diagnosis and treatments using DIS3 mutations.
Reviewer 2 Report
Ohguchi et al., reviewed the characteristics and physiological functions of DIS3 in multiple myeloma (MM). The authors demonstrated the structures and molecular functions of DIS3 as a part of the exosome complex. They highlighted the recent findings of DIS3 functions in RNA homeostasis and the mechanistic insights on MM pathogenesis in regards to DIS3 mutations. The structure of the review is well-organized, and clearly presents the significance of the subject through many details. However, some points need to be improved for publication in IJMS.
- In the introduction or session 2.2, address if DIS3 is widely conserved or not.
- I recommend adding the biological relevance of exosomes at the end of session 2.1 or session 5. Specifically, whether mutation or deletion of other exosome component results in MM or other diseases/cancers.
- In session 2.2, briefly mention DIS3L2 (not considered an exosome component – no need to show as figure).
- End of session 2, are all the DIS3-deficient phenotypes exosome dependent? If not, I recommend rearranging the exosome dependent vs independent defects in DIS3 depletion/mutant.
- Does the transcriptome study show abnormal degradation of the exosomal targets? If the RNA targets are related to MM, it should be mentioned.
- In session 3.1, the authors stated that DIS3-deficient activated B cells exhibit increased DNA:RNA hybrids. The authors also demonstrated that DIS3 removes the ncRNAs to promote chromatin accessibility. Is it specific for activated B cells or other species/tissues? An article mentioned that the DIS3 mutation does not significantly alter R-loop formation (PMID: 30724665 ). Is it due to tissue/species specific? The authors should also discuss this.
- Also, DIS3 is an RNase R that removes overhanging single stranded RNA, whereas DNA:RNA hybrids (R-loops) are removed by RNase H. Does the ncRNA efficiently generate the DNA:RNA hybrid? Are the substrates (DNA:RNA hybrids) for DIS3 different from the R-loops produced during normal transcription? The authors need to address this topic in session 3.
- Making a simple cartoon for session 4.4 will help the readers understand the information better.
- Throughout the manuscript, the authors focus on the DIS3/exosome functions in the nucleus, and also targeting non-coding RNAs. However, the transcriptome study in session 4.4 implies that DIS3 may affect the stability of cytoplasmic RNAs. The authors should address potential RNA (mRNA) targets of DIS3 that can cause MM if DIS3 or exosome component is mutated.
